# Antinociceptive Effect of Hinokinin and Kaurenoic Acid Isolated from *Aristolochia odoratissima* L.

**DOI:** 10.3390/molecules25061454

**Published:** 2020-03-24

**Authors:** Rosa Mariana Montiel-Ruiz, Marcos Córdova-de la Cruz, Manasés González-Cortázar, Alejandro Zamilpa, Abraham Gómez-Rivera, Ricardo López-Rodríguez, Carlos Ernesto Lobato-García, Ever A. Blé-González

**Affiliations:** 1Centro de Investigación Biomédica del Sur (CIBIS), Instituto Mexicano del Seguro Social (IMSS), Argentina No. 1, Col. Centro, Xochitepec 62790, Morelos, Mexico; montielrmariana@gmail.com (R.M.M.-R.); azamilpa_2000@yahoo.com.mx (A.Z.); 2Universidad Juárez Autónoma de Tabasco, Carretera Cunduacán-Jalpa Km. 0.5, Cunduacán 86690, Tabasco, Mexico; marcoscordovaqfb@gmail.com (M.C.-d.l.C.); abgori@gmail.com (A.G.-R.); richard_lorr@hotmail.com (R.L.-R.); Carlos.lobato@ujat.mx (C.E.L.-G.)

**Keywords:** *Aristolochia odoratissima* L., antinociception, hinokinin, kaurenoic acid, formalin test

## Abstract

*Aristolochia odoratissima* L. is employed for the treatment of pain and as an antidote against the poison of venomous animals in traditional medicine. However, reports have not been found, to our knowledge, about the evaluation of the antinociceptive activity of extracts nor about the presence of compounds associated with this activity. Thus, the purpose of this work was to evaluate the antinociceptive activity of extracts and compounds isolated from the stems of *Artistolochia odoratissima* L. The extracts were obtained with solvents of increasing polarity and the compounds were isolated and characterized by column chromatography, HPLC, and NMR. The antinociceptive activity was carried out by the formalin test in mice. Ethyl acetate (AoEA) and methanolic (AoM) extracts decreased the paw licking in both phases of the formalin test. The isolated compounds (kaurenoic acid and hinokinin) from AoEA showed the highest antinociceptive activity in both phases of the formalin test. These results confirmed the analgesic effect of this specie described in traditional medicine and provided a base for a novel analgesic agent. They also allowed an approach for the development of standardized plant extracts with isolated metabolites.

## 1. Introduction

Pain is a complex experience that involves affective, motivational and cognitive aspects, and act as an alarm system to minimize the contact with noxious stimuli [1]. However, when pain persists, it constitutes a public health problem that affects the quality of life of persons who suffer it and therefore, the presence of pain may become a socioeconomic burden. It has been estimated that in the first two decades of the XXI century, more than 5.5 billion people (80% of the world population) live in countries with low or null access to treatment for pain [2]. In addition, studies have shown that approximately 60 million persons suffer from chronic pain. Also, the World Health Organization (WHO) indicates that, each year, one in 10 adults are diagnosed with chronic pain associated with a particular disease, such as rheumatoid arthritis, cancer, diabetes, or post-surgical lesions and procedures [3].

The treatment for this problem is mainly constituted by the administration of non-steroidal anti-inflammatory drugs (NSAIDs), which produce an anti-inflammatory, antipyretic, and analgesic effect and include a broad variety of formulations and doses [4]. NSAIDs have the capacity to block the synthesis of prostaglandins (PGs) by the inhibition of the cyclooxygenase enzymes (COX-1 and COX-2). It is known that COX-1 produces PGs and thromboxane A2, which controls the mucosal barrier in the gastrointestinal tract, renal homeostasis, platelet aggregation, and other physiological functions. While, COX-2 produces PGs related with inflammation, pain, and fever [5]. Another important group of analgesics are opioids, which have the capacity to act on central and peripheral sites, they inhibit pain pathways by binding with opioid receptors [6], which are widely distributed in the central and peripheral nervous system, and in the gastrointestinal tract [7]. However, the continuous use of analgesics generates adverse effects, such as gastrointestinal damage, renal toxicity, tolerance, respiratory depression, and physical dependence [8,9,10]. Thus, the search for novel and effective analgesic compounds constitutes a challenge. In this context, pharmacological studies have been conducted on the extracts of plants employed in traditional medicine. This research have led to the isolation of compounds which have been identified to exhibit both antinociceptive and anti-inflammatory effects [11]. For example, phytochemical analysis furofuran lignans isolated from *Pleurothyrium cinereum* and *Ocotea macrophylla* exert a selective inhibitory activity against COX-1 and COX-2; similarly terpenes such as lemnalol and madecassoside isolated from several plant species have shown a selective activity against COX-2 [12,13].

*Aristolochia odoratissima* L. (Aristolochiaceae family) is commonly known as *cocoba* or *guaco* in Tabasco, Mexico. It is a climber plant that is distributed in tropical and subtropical climates in Mexico. *Aristolochia odoratissima* is employed for the treatment of pain and as an antidote against snake venom in traditional medicine [14]. The parts used are the stem and the rhizome as an infusion in water. Chemical study of the essential oil of the leaf of this plant reported the presence of a high proportion of sesquiterpenes and, in minor proportion, mono- and diterpenes [15]. To date, a study has confirmed the effect of *Aristolochia odoratissima* against snake venom and the isolation of (−)-cubebin and aristolochic acid from the essential oil [16,17]. To our knowledge, phytochemical analysis and the antinociceptive activity of *Aristolochia odoratissima* has not been investigated. Therefore, the present study was undertaken to demonstrate the antinociceptive effect of extracts and isolated compounds obtained from the stem of *Aristolochia odoratissima* using chemical induced nociception in mice.

## 2. Results

### 2.1. Antinociceptive Effect of the Extracts of Aristolochia odoratissima

We employed 800 g of dry and ground material for the extraction procedure and we obtained the following amounts and percentages of each of the extracts: AoEA (17 g, 2.12%); AoM (22 g, 2.75%), and AoA (20 g, 2.5%). Figure 1 shows the antinociceptive effect of AoEA extract in formalin tests. In the early phase, formalin administration in the right hind paw of mice produced nociceptive responses of licking and biting of the injected paw. Administration of AoEA extract (100 mg/kg, i.p.) produced a significant (*p* < 0.001) inhibition of the nociceptive response (41%). This effect was higher than the one produced by the administration of 30 mg/kg of diclofenac (15%) and it was similar to the effect produced by 5 mg/kg of tramadol (42%). In the late phase, the effect of the AoEA extract in nociceptive response was significant (*p* < 0.05 and 0.001) at a dose rate of 30 (55%) and 100 (66%) mg/kg. The effect of diclofenac (30 mg/kg) and tramadol (5 mg/kg) were 58% and 72%, respectively.

The antinociceptive effect generated by the AoM extract is presented in Figure 2. In phase 1, the 30 and 100 mg/kg doses significantly inhibited pain with percentages values of 31% and 28%, respectively. However, in phase 2, the 30 mg/kg dose did not have a significant effect. At 30 and 100 mg/kg dose of AoM, the antinociceptive effect was 48 and 42%, respectively, these values have a significant difference when compared to the control group (DMSO, 3%) but their effect is lower than the reference drugs.

The effect of the aqueous extract (AoA) is presented in Figure 3. Both the 30 and 100 mg/kg doses of AoA extract did not show significant difference when compared to the vehicle group in both phases of the formalin test in mice.

### 2.2. Antinociceptive of Kaurenoic Acid (**1**) and Hinokinin (**2**) Isolated from Aristolochia odoratissima

The administration of kaurenoic acid and hinokinin at a dose rate of 15 mg/kg (i.p.) produced a significant (*p* < 0.001) antinociceptive effect compared to the vehicle group (DMSO, 3%) in both early and late phases of the experiment. Kaurenoic acid and hinokinin decreased by 38% and 48%, respectively, the paw licking time in early phase, as well as 69% and 82% respectively, in the late phase of the formalin test in mice. These results are higher than the effect shown by both diclofenac and tramadol (Figure 4).

### 2.3. Characterization by NMR of the Compounds (**1–3**)

Compound **1** was obtained as dichloromethane-soluble colorless crystals. The ^13^C NMR spectrum of **1**, showed 20 characteristic signals for a diterpene structure and according to the DEPT experiment, five signals correspond to quaternary carbons, three correspond to methines, ten are assigned as methylenes, and there are two methyl groups. The ^1^H NMR spectrum exhibited a double-linked AB system: [δ 4.81 (1H, s), 4.75 (1H, s)] (For spectra see Appendix A). This evidence and the comparison of the spectroscopic data with those described in the literature [18] lead us to indicate that this compound corresponds to *ent*-kaur-16-en-19-oic acid, commonly known as kaurenic acid (**1**) (Figure 5).

*Compound* (**1**): ^1^H-NMR (600 MHz, CDCl_3_): δ 4.75 (1H, s, br, H-17a), 4.81 (1H, s, br, H-17b), 0.96 (3H, s, H-20), 1.25 (3H, s, H-18); ^13^C-NMR (150 MHz, CDCl_3_): δ 40.7 (C-1), 19.1 (C-2), 37.8 (C-3), 43.7 (C-4), 57.0 (C-5), 21.8 (C-6), 41.2 (C-7), 44.2 (C-8), 55.1 (C-9), 39.6 (C-10), 18.4 (C-11), 33.1 (C-12), 43.8 (C-13), 39.7 (C-14), 48.9 (C-15), 155.8 (C-16), 102.9 (C-17), 28.9 (C-18), 184.2 (C-19), 15.5 (C-20). 

Compound **2** was obtained as a dichloromethane-soluble yellow oil. The ^13^C NMR spectrum of **2** showed 20 signals: Seven quaternary carbons; eight methines and four methylenes (identified by the DEPT experiment). The ^1^H NMR spectrum of **2**, displayed two ABX aromatic-ring systems: one in [δ 6.63 (1H, d, 1.4 Hz, H-2), 6.73 (1H, d, 7.7 Hz, H-5) and 6.60 (1H, dd, 1.4, 7.7 Hz, H-6)], and the other in [δ 6.47 (1H, d, 1.8 Hz, H-2′), 6.7 (1H, d, 8.4 Hz, H-5′) and 6.47 (1H, dd, 2.2, 8 Hz, H-6′)]. According to the HMBC experiment they are correlated with a methylenedioxy group (-O-CH_2_-O-) located in δ 5.93 (2H, m) and with two signals in δ 3.0 (1H, dd, 5.1, 14.3 Hz, H-7a) and 2.84 (1H, dd, 7.3, 14.3 Hz, H-7b), which indicates the presence of a benzyl fragment that is attached to a tetrahydrofuran ring, resulting in a lignan-type skeleton (For spectra see Appendix A). According to these spectroscopic analysis (Table 1) and the comparison with the data described in the literature [19,20] lead us to indicate that this compound corresponds to (*3R*, *4R*)-3,4-bis(1,3-benzodioxol-5-ylmethyl)oxolan-2-one (**2**), known as hinokinin (Figure 6).

Compound **2**: ^1^H-NMR (600 MHz, CDCl_3_) and ^13^C-NMR (150 MHz, CDCl_3_) (see Table 1).

Compound **3** was obtained as an oil. Analysis of the ^1^H and ^13^C NMR spectra, exhibited similarities with the spectroscopic signals found for compound **2**. However, we found the presence of a ketal group in δ 104.6, which was attributed to the reduction of the carbonyl group of the lactone in C-9. This structural feature was supported by the observation in the proton spectrum of a signal in δ 5.14 (1H, d, 1.6 Hz, H-9) which possesses a HMBC correlation with a C-O signal in δ 72.3 assigned to C-9′. Analysis of the spectra ((For spectra see Appendix A).) allowed us to determine that this compound corresponds to the 8,8′ *trans*-cubebin (**3**), known as cubebin (Figure 7).

Compound **3**: ^1^H-NMR (600 MHz, CDCl_3_) and ^13^C-NMR (150 MHz, CDCl_3_) (see Table 1) [19,20].

## 3. Discussion

*Aristolochia odoratissima* is widely used in Mexican traditional medicine for the treatment of pain and as an antidote against snake venom. In this work, we evaluated the antinociceptive activity of the extracts and compounds isolated from the stem of *Aristolochia odoratissima* using a chemically induced pain test in mice. The formalin test, a method commonly employed to study the antinociceptive activity of drugs, has been reported to produce a characteristic biphasic nociceptive response [21,22]. The early or neurogenic phase represents pain centrally mediated, evoked by the direct stimulation of the formalin in the primary afferent C-fibers, that results in the release of substance P, serotonin, or and/or CGRP (calcitonin gene-related peptide). Meanwhile, the late or inflammatory phase, reflects pain peripherally mediated, that is primarily due to an inflammatory reaction in the peripheral tissue mediated by the release of various mediators such as PG‘s, COX, and nitric oxide (NO), as well as the sensitizing of the neurons of the dorsal horn of the spinal cord [21] Centrally acting drugs, such as opioids, inhibit both phases of pain, while peripherally acting drugs, such as diclofenac, only inhibits pain during the late phase [22,23].

The results of the present study revealed that the intraperitoneal (i.p.) administration of the AoEA and AoM extracts of the *Aristolochia odoratissima* stems possesses an antinociceptive effect in the formalin test. AoEA showed the greatest effect in both early and late phases, suggesting that the extract contains metabolites with the capacity to inhibit nociception caused by the formalin at the peripheral and central level. Therefore, the mechanism of action of AoEA could be related to the inhibition of COX-1 and/or COX-2 and other derived inflammatory mediators including histamine and bradykinin. In addition, the effect of AoEA may involve the inhibition of the activation of the nociceptor present on afferent C-fiber as well as the release of glutamate and substance P.

The presence of lignans in *Aristolochia* species has been previously described; and a previous work has already reported the presence of (−)-cubebin in the essential oil of *Aristolochia odoratissima* [24,25,26,27,28]. However, this research presents for the first time (to our knowledge) the presence of kaurenic acid and hinokinin in this medicinal plant. Both kaurenic acid and hinokinin produced the greatest antinociceptive activity at the highest dose evaluated in the formalin test. Previous studies have reported the analgesic activity of kaurenoic acid, a diterpene, in thermal-induced pain (tail-flick test), as well as in chemical-induced pain such as the intraplantar administration of capsaicin [29]. The mechanism of action of kaurenoic acid is by inhibiting proinflammatory cytokines and the activation of the NO-cyclic GMP-ATP-K^+^ channels signaling pathway [30,31]. Other biological activities that have been reported for this diterpene include anti-inflammatory, hypotensor, and diuretic effects in in-vivo models, as well as a smooth-muscle relaxant and cytotoxic activity in in-vitro models [32,33].

Hinokinin showed the greatest antinoceptive activity in both phases of the formalin test. The results indicate that this compound is able to produce both central and peripheral effects. In terms of the mechanism of action, it has been demonstrated by in vitro assays that hinokinin and synthetic derivatives selectively inhibit COX-1 and COX-2 [34,35,36]. Furthermore, previous studies have reported that this compound has antitumoral, anti-inflammatory, antimicrobial and anti-trypanosomal activity [34].

Based on these data, this study supports the antinociceptive effect of *Aristolochia odoratissima*, which is employed in traditional medicine. Furthermore, the presence of metabolites such as kaurenic acid and hinokinin could be related to the biological activity.

## 4. Materials and Methods

### 4.1. Plant Material

The stems of *Aristolochia odoratissima* were collected on May 2016, in Poblado C-9 Francisco I. Madero, Cárdenas, Tabasco, Mexico (18°9′46.84″ N; 93°28′14.88″ O; 7.0 m.a.s.l.). A specimen was deposited at the Herbarium of Academic Division of Biological Sciences of the Juárez Autonomous University of Tabasco for its taxonomic identification (Voucher No. 036502).

### 4.2. Extraction and Isolation

The stems of *Aristolochia odoratissima* were dried at room temperature for 72 h and grounded to a 4 mm particle size (Pulvex brand, Mexico City, Mexico). The dry plant material (800 g) was successively extracted three times with ethyl acetate (2.0 L, Merck, Mexico City, Mexico) at room temperature, after each extraction, the solvent was filtered and vacuum-concentrated at 40 °C in a rotary evaporator (Heidolph G3, Heidolph, Schwabach, Germany) to obtain the ethyl acetate extract (AoEA). To obtain the methanolic (AoM) and aqueous (AoA) extracts, the plant’s dry residue was macerated with methanol (2.0 L; Merck, Mexico City, Mexico) and later with water (2.0 L; Merck, Mexico City, Mexico) following the same procedure previously described. All the extracts were lyophilized (Heto Drywinner DW3, ThermoFisher SCIENTIFIC, Mexico City, Mexico).

### 4.3. Isolation and Identification of the Compounds

The AoEA extract (7 g) was adsorbed onto silica gel (7 g, Gel 60, Merck, Mexico City, Mexico) and placed in a silica gel-packed glass column (600 × 6050 mm) (100 g, 70–230 mesh, Merck, Mexico City, Mexico) as stationary phase and, as mobile phase, a mixture of *n*-ethyl hexane-acetate was employed with an increase in polarity of 5% *v*/*v*, collecting 33 fractions of 200 mL each. In the AoEA-3, AoEA-9, and AoEA-11 fractions, only one TLC point was shown, and it was identified as kaurenoic acid (**1**; 40 mg), hinokinin (**2**; 35 mg), and cubebin (**3**, 12 mg), respectively. Only compounds **1** and **2** were evaluated in the biological model. The chemical characterization of these structures was carried out by means of the analysis of the 1D and 2D NMR spectra. The spectra were obtained in a BrukerAdvance III HD-600 (600 MHz, Bruker, Fällanden, Switserland,) equipment and the solvent employed was CDCl_3_ for the three compounds. The chemical displacements are reported in ppm, while tetramethylsilane (TMS) was employed as internal standard.

### 4.4. Experimentation Animals

Male ICR mice with a weight range of 25–30 g, from Envigo RMS S.A. de C.V. (Mexico City, Mexico), were used throughout the experiments. These animals were maintained in the Bioterium of Centro de Investigación Biomédica del Sur (CIBIS-IMSS) under 12 h–12 h light-dark cycle and constant temperature (23–25 °C) with free access to food and water. The experimental protocol was approved by the Institutional Commission for Ethics in Research of the Juárez Autonomous University of Tabasco, and received the registration number DI/524/2020. The animals were treated under the Mexican federal regulations (Norma Oficial Mexicana for care and use of laboratory animals, NOM-062-ZOO-1999) and the ethical guidelines for the investigation of pain in conscious animals [37]. The number of animals and the intensity of the noxious stimuli employed were the minimum necessary to demonstrate the consistent effects of the pharmacological treatments. Mice were used only once during the experiments and were euthanized in a CO_2_ chamber immediately after the experiment.

### 4.5. Drugs and Reagents

The drugs and reagents employed were acquired from Sigma-Aldrich (Toluca, México): formaldehyde; dimethylsulfoxide (DMSO), diclofenac (Dic), and tramadol (Tra). For the evaluation of the antinociceptive activity, diclofenac (30 mg/kg, i.p.) and tramadol (5 mg/kg, i.p.) were used as positive groups and DMSO (3%, diluted in water) as a negative group.

### 4.6. The Formalin Test

The method used for this test was similar to that described previously by Dubuisson and Dennis [38] and modified by Hunskaar and Hole [23]. This experimental test consists of placing each mouse in an open observation chamber to allow them to accommodate to their surroundings. Each mouse was removed for subcutaneous (s.c.) administration of 25 µL of diluted 2% formalin into the right hindpaw with a 30-gauge needle. The time of licking and biting of the injected paw was defined as a nociceptive response, which was recorded in periods of 5 min during 30 min after the analgesic injection [23]. Nociceptive response is biphasic: the initial (early phase, 0–5 min) results by the direct stimulation of nociceptors, it is of rapid onset and great intensity. The second (late phase, 15–30 min) is secondary to inflammatory reactions [22,23]. Groups of five animals were intraperitoneally (i.p.) administered with: Vehicle (DMSO, 3%), diclofenac (30 mg/kg), tramadol (5 mg/kg), extracts (30 and 100 mg/kg), or compounds (5 and 15 mg/kg), 30 min before to the analgesic stimuli. Doses were chosen by pilot tests and previous studies [39,40,41].

### 4.7. Statistical Analysis

The data are expressed as the mean ± standard error of the mean (SEM), and statistical significance was determined using an analysis of variance (ANOVA) + Dunnett´s test. Values were considered significant different at ** p* < 0.05 and *** p* < 0.01. All analysis were performed using by GraphPad Prism ver. 7.0 statistical program (Graph pad Prism version 7.05 for windows, GraphPad Software, La Jolla, CA, USA, 2018).

## 5. Conclusions

The present study demonstrated the antinociceptive potential of the extracts and compounds isolated from the *A*. *odoratissima* stem, mainly from the AoEA extract and hinokinin, which were active in both phases of the formalin model. Therefore, we conclude that the stems of this plant can be employed as a therapeutic alternative in the treatment of pain. The results obtained permit the confirmation of the analgesic use of this species in traditional medicine and provide a base for new studies on antinociceptive activity, as well as the development of standardized extracts with isolated metabolites.

## Figures and Tables

**Figure 1 molecules-25-01454-f001:**
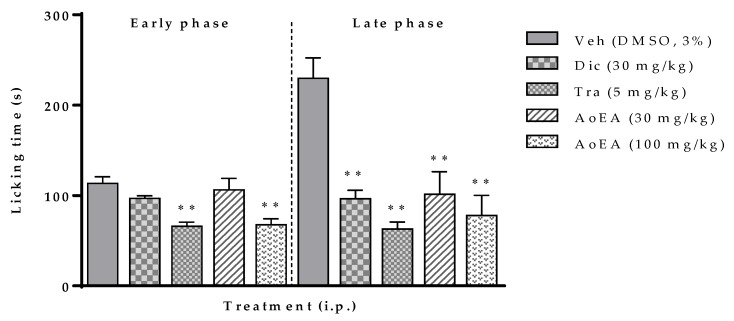
Effect of the ethyl acetate extract of *Aristolochia odoratissima* (AoEA, 30 and 100 mg/kg, i.p.) in the formalin test (2.5%, 25 µL/paw) in mice. The data are expressed with the mean ± sem. One-way ANOVA + Dunnett test, * *p* < 0.05 and ** *p* < 0.001. Veh: Vehicle; Dic: Diclofenac, Tra: Tramadol.

**Figure 2 molecules-25-01454-f002:**
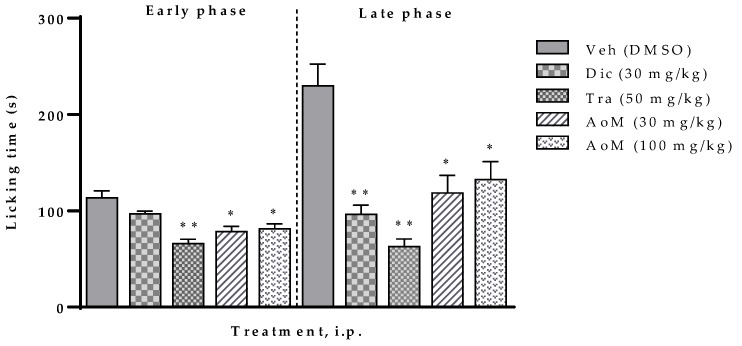
Effect of the methanolic extract of *Aristolochia odoratissima* (AoM, 30 and 100 mg/kg, i.p.) in the formalin test (2.5%, 25 µL/paw) in mice. The data are expressed with the mean ± sem. One-way ANOVA + Dunnett test, * *p* < 0.05 and ** *p* < 0.001. Veh: Vehicle; Dic: Diclofenac, Tra: Tramadol.

**Figure 3 molecules-25-01454-f003:**
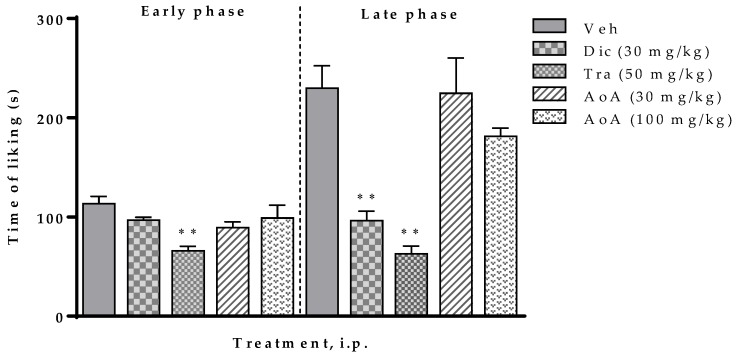
Effect of the aqueous extract of *Aristolochia odoratissima* (AoA, 30 and 100 mg/kg, i.p.) in the formalin test (2.5%, 25 µL/paw) in mice. The data are expressed with the mean ± sem. One-way ANOVA + Dunnett test, * *p* < 0.05 and ** *p* < 0.001. Veh: Vehicle; Dic: Diclofenac, Tra: Tramadol.

**Figure 4 molecules-25-01454-f004:**
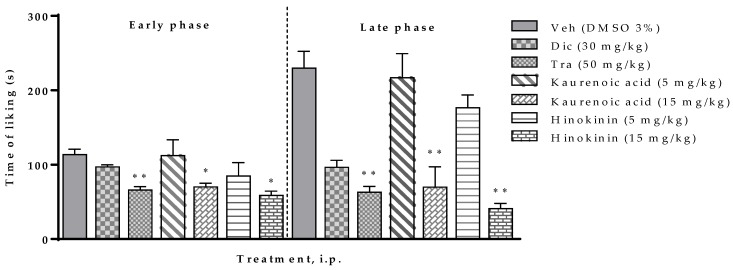
Effect of the kaurenoic acid and hinokinin (5 and 15 mg/kg, i.p.) from *Aristolochia odoratissima* in the formalin test (2.5%, 25 µL/paw) in mice. The data are expressed with the mean ± sem. One-way ANOVA + Dunnett test, * *p* < 0.05 and ** *p* < 0.001. Veh: Vehicle; Dic: Diclofenac, Tra: Tramadol.

**Figure 5 molecules-25-01454-f005:**
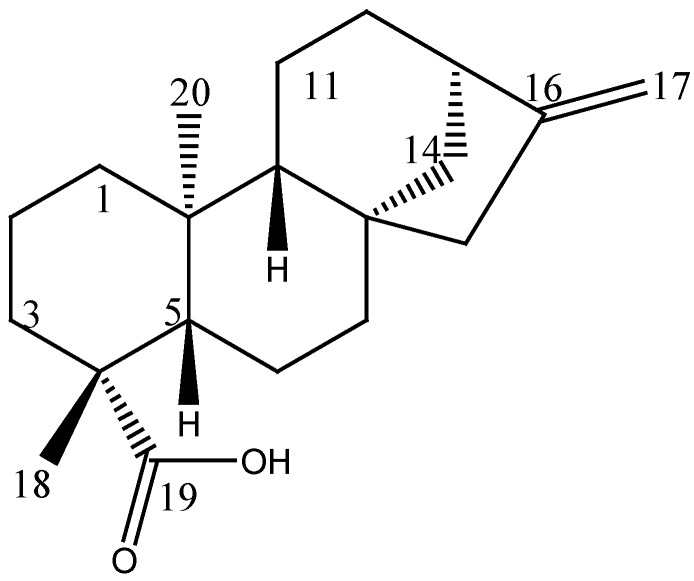
Chemical structure of kaurenoic acid (**1**).

**Figure 6 molecules-25-01454-f006:**
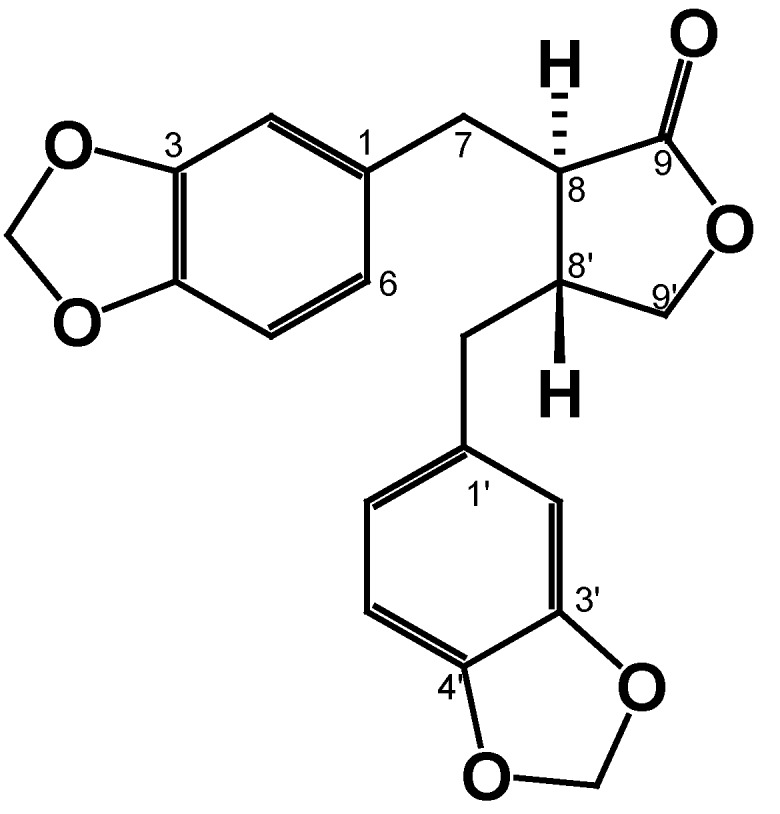
Chemical structure of hinokinin (**2**)**.**

**Figure 7 molecules-25-01454-f007:**
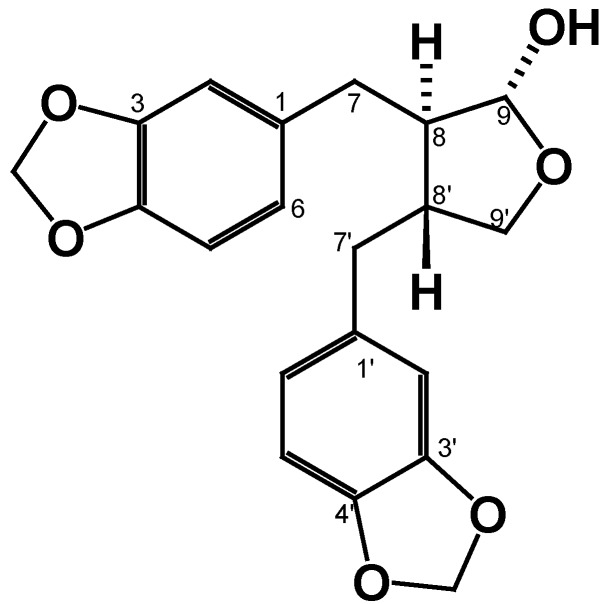
Chemical structure of 8,8′ trans-cubebin (**3**)**.**

**Table 1 molecules-25-01454-t001:** NMR spectroscopy data (^1^H and ^13^C, in CDCl_3_) of compounds **2** and **3.**

Position	δ ^13^C 2	δ ^1^_H_ (*J* in Hz) 2	δ ^13^C 3	δ ^1^_H_ (*J* in Hz) 3
1	131.5		134.6	
2	109.3	6.63 (1H, d, 1.4)	109.5	6.58 (1H, s, br)
3	147.8		147.9	
4	146.4		146.1	
5	108.2	6.73 (1H, d, 7.7)	108.3	6.69 (1H, d, 7.7)
6	122.1	6.60 (1H, dd, 1.4, 7.7)	122.1	6.55 (1H, d, br, 7.7)
7	38.3	a 3.0 (1H, dd, 5.1, 14.3) b 2.84 (1H, dd, 7.3, 14.3)	39.3	a 2.53 (1H, dd, 6.2, 13.9) b 2.53 (1H, dd, 7.3, 13.5)
8	46.4	2.53 (1H, ddd, 5.1,7.7, 8.0)	52.6	2.12 (1H, m)
9	178.3		104.6	5.14 (1H, d, 1.6)
1′	131.2		133.8	
2′	108.7	6.47 (1H, d, 1.8)	109.2	6.51 (1H, s, br)
3′	147.8		147.9	
4′	146.2		146.1	
5′	108.1	6.7 (1H, d, 8.4)	108.4	6.69 (1H, d, 7.7)
6′	121.4	6.47 (1H, dd, 2.2, 8)	121.7	6.50 (1H, d, br, 8.0)
7′	34.7	a 2.60 (1H, dd, 9.9, 17.2) b 2.47 (1H, dd, 8.4, 16.8)	38.7	a 2.64 (1H, dd, 7.7, 13.9) b 2.43 (1H, dd, 7.3, 13.5)
8′	41.2	2.46 (1H, m)	46.0	2.12 (1H, m)
9′	71.0	a 4.13 (1H, dd, 6.9, 9.5) b 3.86 (1H, dd, 7.3, 9.1)	72.3	a 3.92 (1H, dd, 7.3, 8.4) b 3.41 (1H, dd, 8.0, 8.4)
1-O-CH_2_-O	100.9	5.93 (2H, m)	101.1	5.93 (2H, m)
2-O-CH_2_-O	100.9	5.93 (2H, m)	101.1	5.90 (2H, m)

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
