# Peer review of "Antinociceptive Effect of Hinokinin and Kaurenoic Acid Isolated from *Aristolochia odoratissima* L."

_molecules, 2020, doi:10.3390/molecules25061454_

Round 1

Reviewer 1 Report

The manuscript is well-written and the results support the conclusions. Some corrections and clarifications should addressed before publication:

Line 22: Please correct "extract" in "extracts".

Please include the GPS coordinates for plant harvesting.

Please indicate the extraction temperature for maceration.

Please indicate the number of animals used for experimental procedure.

Additionally, please indicate the criteria for choosing the employed dosages.

Author Response

Reviewer 1: The manuscript is well-written and the results support the conclusions. Some corrections and clarifications should addressed before publication:

Line 22: Please correct "extract" in "extracts".

DONE

Please include the GPS coordinates for plant harvesting.

In 4.1 section (204) was written: (18°9’46.84´´ N; 93°28´14.88´´O;  7.0 m.a.s.l.)

Please indicate the extraction temperature for maceration.

In 4.2 section (208) was written:  at room temperature

Please indicate the number of animals used for experimental procedure.

In 4.6 section (252) we rewrite as follows: Groups of five animals were intraperitoneally (i.p) administered with…

Additionally, please indicate the criteria for choosing the employed dosages.

Line 253-254 was written: Doses were chosen by pilot tests and previous studies [39-41].

Reviewer 2 Report

A paper describing the analgesic effects of extracts from a plant used for traditional medicine in Mexico with reported analgesic properties. The paper is generally well written however moderate english language revision is required throughout. Some specific comments include the following;

Was animal ethics committe approval obtained to perform this work? Please provide details.

Line 30-32 could be worded better

Line 35 million or billion?

Line 42 NSAIDs have not has

Line45-46 could mention selective COX inhibition by compounds.

Line 49-50 what about opioid dependence

Line 55 tropical and subtropical climates

Line 58 a high proportion

Line 62 species for either of the compounds

Line 77 at a dose rate of 30…..

Line 86 did not have a significant effect

Line 95-95 language editing

Line 104 at a dose rate of

Line 108 than that shown by

Line 160 using a chemically induced pain test not on chemical

Line 174 related to not related with

Line 176 in the inhibition of

Line 178-180 rewrite this sentence

Line 189-190 rewrite

Line 225 how many mice were used

Line 230 how was number determined? How was intensity determined?

Author Response

Reviewer 2: A paper describing the analgesic effects of extracts from a plant used for traditional medicine in Mexico with reported analgesic properties. The paper is generally well written however moderate english language revision is required throughout. Some specific comments include the following

Was animal ethics committe approval obtained to perform this work? Please provide details.

The experimental protocol was approved by the Institutional Commission for Ethics in Research of the Juárez Autonomous University of Tabasco, Mexico, and received registration number DI/524/2020.

Line 30-32 could be worded better

Sentence of line 30-32 was rewritten as follows: Pain is a complex experience that involves affective, motivational and cognitive aspects, and act as an alarm system to minimize the contact with noxious stimuli.

Line 35 million or billion?

Line 35: 5.5 billion people

Line 42 NSAIDs have not has….

DONE

Line 45-46 could mention selective COX inhibition by compounds.

Line 53-56 was written as follows: For example, neo and furofuran lignans isolated of Pleurothyrium cinereum and Ocotea macrophylla exert an selective inhibitory activities againts COX-1 and COX-2; similarly terpenes such as lemnalol and madecassoside isolated from plant species has been showed selective activity against COX-2 [12-13].

Line 49-50 what about opioid dependence

As the reviewer mentioned, opioid dependence is a serious adverse effect. This was also included in the list

Line 55 tropical and subtropical climates

DONE

Line 58 a high proportion

DONE

Line 62 species for either of the compounds

Sentence of line 62 was rewritten as follows:

To our knowledge, phytochemical analysis and the antinociceptive activity of Aristolochia odoratissima has not been investigated.

Line 77 at a dose rate of 30…..

DONE

Line 86 did not have a significant effect

DONE

Line 95-95 language editing

DONE

Line 104 at a dose rate of

DONE

Line 108 than that shown by

DONE

Line 160 using a chemically induced pain test not on chemical

DONE

Line 174 related to not related with

DONE

Line 176 in the inhibition of

DONE

Line 178-180 rewrite this sentence

Line 182-183 was rewritten as follows: The presence of lignans in Aristolochia species has been previously described; and a previous work has already reported the presence of (-)-cubebin in the essential oil of A. odoratissima [26, 29-30]. However, this research presents for the first time (to our knowledge) the presence of kaurenic acid and hinokinin in this medicinal plant.

Line 189-190 rewrite

Line 193-197 was rewritten as follows:

Hinokinin showed the greatest antinoceptive activity in both phases of the formalin test. The results indicate that this compound is able to produce both central and peripheral effects. In terms of the mechanism of action, it has been demonstrated by in vitro assays that hinokinin and synthetic derivatives selectively inhibit COX-1 and COX-2 [35, 36]. Furthermore, previous studies have reported that this compound has antitumoral, anti-inflammatory, antimicrobial and anti-trypanosomal activity [37].

Line 225 how many mice were used

In 4.6 section (252) we rewrite as follows: Groups of five animals were intraperitoneally (i.p) administered with….

Line 230 how was number determined? How was intensity determined?

The number of animals was determined according to the literature where n ≥ 5 animals per group are employed; and the intensity was determined based on the concentration of the formalin previously standardized in our laboratory for this model.